# The Emerging Roles of Antioxidant Enzymes by Dietary Phytochemicals in Vascular Diseases

**DOI:** 10.3390/life11030199

**Published:** 2021-03-04

**Authors:** Seung Eun Lee, Yong Seek Park

**Affiliations:** Department of Microbiology, School of Medicine, Kyung Hee University, Seoul 02447, Korea; eunlee@khu.ac.kr

**Keywords:** vascular diseases, antioxidant enzymes, bioactive compounds

## Abstract

Vascular diseases are major causes of death worldwide, causing pathologies including diabetes, atherosclerosis, and chronic obstructive pulmonary disease (COPD). Exposure of the vascular system to a variety of stressors and inducers has been implicated in the development of various human diseases, including chronic inflammatory diseases. In the vascular wall, antioxidant enzymes form the first line of defense against oxidative stress. Recently, extensive research into the beneficial effects of phytochemicals has been conducted; phytochemicals are found in commonly used spices, fruits, and herbs, and are used to prevent various pathologic conditions, including vascular diseases. The present review aims to highlight the effects of dietary phytochemicals role on antioxidant enzymes in vascular diseases.

## 1. Introduction

Vascular diseases are responsible for numerous deaths annually worldwide. The pathogenesis of vascular disease involves the activation of pro-inflammatory signaling pathways, expression of cytokines/chemokines, and elevated oxidative stress. Exposure to oxidative stress may directly injure the vasculature and induce vascular dysfunction by producing dysregulation of the immune response. Oxidative stress is caused by an imbalance between the production and accumulation of reactive oxygen species (ROS) and the capacity of antioxidant defense mechanisms favoring oxidants [1]. ROS generation may lead to cellular necrosis by damaging the mitochondria and stimulating pro-apoptotic signaling. ROS play a vital role in the progressive pathology of vascular diseases, including inflammatory responses, apoptosis, cell growth, and endothelial dysfunction [2,3]. Hence, strategies that target oxidative stress may have enormous therapeutic potential for the prevention of vascular diseases. The oxidative balance in the vasculature is tightly regulated by a wealth of pro- and antioxidant systems that contain antioxidant enzymes. A better understanding of oxidative stress and modulation of antioxidant enzymes is necessary for the broader use of pharmacological and regenerative therapies for vascular diseases. Here, we summarize the critical roles of antioxidant enzymes in vascular diseases and discuss the potential therapeutic roles of phytochemicals that target antioxidant enzymes.

## 2. Role of Antioxidant Enzymes

Escalated oxidative stress has been implicated in numerous human diseases, and is associated with disease severity. Vascular structures contain antioxidant defense systems that scavenge ROS [4]. Antioxidant enzymes are important as part of the cellular defense mechanism against free radical generation and in the prevention and repair of free-radical-produced molecular damage in a variety of situations [5]. Under physiological and pathophysiological conditions, various enzymes are responsible for modulating redox balance. Thus, antioxidant enzymes are vital for the maintaining the homeostasis of oxidants. Such antioxidant enzymes include catalase (CAT), glutathione reductase (GR), thioredoxin reductase (TrxR), heme oxygenase-1 (HO-1), superoxide dismutase (SOD), glutathione peroxidase (GPx), peroxiredoxin (Prx), paraoxonases (PON), and NAD(P)H: quinone oxidoreductase 1 (NQO1) (Figure 1).

### 2.1. Catalase (CAT)

CAT, a heme-containing tetrameric protein, is a normal component of cellular peroxisomes. CAT converts hydrogen peroxide (H_2_O_2_) into water and molecular oxygen and protects cells against excessive ROS production as well as inhibiting the accumulation of H_2_O_2_. CAT is universally expressed but is primarily located in the peroxisomes of all types of mammalian cells [6] and human vascular cells [7]. CAT is also universally expressed by neurons and glial cells in the central nervous system. Altered antioxidant enzyme activity, including decreased catalase activity in neutrophils, has been described [8,9]. Parastatidis et al. demonstrated that altered H_2_O_2_ signaling as a result of reduced CAT activity in the vasculature could be an early insult that leads to aortic dilatation [10]. Similarly, Wang et al. reported the anti-atherosclerotic effect of pterostilbene (a natural dietary compound) on experimental atherosclerosis via regulation of CAT/PTEN signaling [11].

### 2.2. Glutathione Reductase (GR)

Glutathione plays a fundamental antioxidant intracellular role [12] and is implicated in several metabolic processes. Oxidized glutathione molecules form disulfide bonds with each other until they are converted to reduced glutathione by GR. GR is a crucial enzyme in gene regulation, the maintenance of high rates of reduced glutathione (GSH)/oxidized disulfide (GSSG), intracellular signal transduction, elimination of free radicals and reactive oxygen species, and the preservation of intracellular redox status [13].

### 2.3. Thioredoxin Reductase (TrxR)

The thioredoxin system consists of thioredoxin (Trx), thioredoxin reductase (TrxR), and nicotinamide adenine dinucleotide phosphate (NADPH), which mainly regulates intracellular redox homeostasis [14]. The three mammalian genes encode different TrxR isoforms. In mice, Txnrd1 encodes cytosolic TrxR1, Txnrd2 encodes mitochondrial TrxR2 (also called TR3), and Txnrd3 encodes thioredoxin glutathione reductase (TGR), which is primarily expressed in the spermatids of the testis and seems to play a significant role in spermatogenesis [15,16]. 

### 2.4. Heme Oxygenase-1 (HO-1)

HO is the first, rate-limiting enzyme in the catalysis of cellular heme degradation and the production of carbon monoxide, biliverdin, and free iron [17]. Two functional HO isoforms have been identified in mammals: HO-1 and HO-2 [18,19]. HO-1 is ubiquitously distributed and highly inducible by a wide variety of inducers, including endotoxins, metals, oxidants, cytokines, and phytochemicals [20,21,22]. There is ample evidence to suggest that HO-1 can protect against vascular remodeling and atherogenesis [23].

### 2.5. Superoxide Dismutase (SOD)

SODs are widespread and are primary regulatory enzymes used by microorganisms to catalyze the conversion of superoxide into oxygen and H_2_O_2_ [24]. Mammals express three SODs isoforms: the cytoplasmic copper/zinc-dependent SOD (SOD1; Cu/ZnSOD is a 32 kDa homodimeric enzyme), the mitochondrial manganese-dependent SOD (SOD2; MnSOD is an 88 kDa homotetrameric enzyme), and the extracellular Cu/ZnSOD (SOD3; is also dependent upon copper and zinc, contains a signaling peptide sequence, and exists as a homotetramer of 135 kDa) [25,26]. Several studies have reported a relationship between SOD3 and pathological conditions that involve vascular dysfunction, such as diabetes and cardiovascular disease [26,27].

### 2.6. Glutathione Peroxidase (GPx)

The GPx family comprises three evolutionary groups arising from a Cys-containing ancestor that catalyzes the reduction of H_2_O_2_. Eight isoforms of GPx have been identified, of which five are selenoproteins (GPx1-4 and GPx6). The three selenium-independent GPx enzymes rely on thiol rather than selenol chemistry. Among the GPx enzymes, GPx1 is the most abundant and ubiquitous isoform, whereas GPx6 is found as a selenoprotein only in humans [28]. The anti-atherosclerotic properties of GPx-1 in endothelial cells [29] and an inhibitory potential against lipid peroxidation of GPx-4 in mice have been reported [30]. GPx2 was first found in gastrointestinal tissues, and GPx3 is mainly synthesized in the proximal convoluted tubule cells of the kidney.

### 2.7. Peroxiredoxin (Px)

Prxs are a family of thiol-dependent peroxidases that neutralize reactive oxygen/nitrogen species and protect against oxidative and inflammatory stress [31]. Prx enzymes are able to form disulfide linkages following oxidation, and return to their active form following reduction by an additional enzyme, thioredoxin [32]. Six mammalian Prx isozymes (Prx1-6) have been classified based on the mechanism and the number of cysteine residues involved during catalysis [33]. Six Prx isoforms are expressed in mammals: the cytosolic isoforms Prx1, Prx2, and Prx6; the mitochondrial isoform Prx3; the secreted isoform Prx4; and Prx5, which is localized in multiple organelles [34]. Schreibelt et al. suggested that vascular Prx1 functions as an endogenous defense mechanism [35]. Guo et al. found that Prx4 is an anti-atherogenic factor that suppresses oxidative damage and apoptosis [36].

### 2.8. Paraoxonase (PON)

There are three different isoforms of PON: PON1, PON2, and PON3, which have multifunctional roles in numerous biological pathways, including protection against oxidative damage [37]. Among these, increasing attention has been focused on the role of PON1, which is a key functional constituent of high-density lipoprotein (HDL) particles in various human diseases, including diabetes, cardiovascular disease, cancers, aging, and several neurological disorders [38]. PON2 is an intracellular enzyme with antioxidant effects in major vascular cells [39]. Mechanistically, PON3 seems similar to PON2 [40], and transgenic PON3 expression lowers atherosclerosis and adiposity [41].

### 2.9. NAD (P) H: Quinone Oxidoreductase 1 (NQO1)

NQO1 is a broadly distributed FAD-dependent flavoprotein that stimulates obligatory two-electron reductions of various exogenous and endogenous quinones, quinoneimines, nitroaromatics, and azo dyes [42]. These reductions decrease quinone levels, thereby diminishing the chances of producing reactive oxygen intermediates by redox cycling and for the attenuation of intracellular thiol pools. Several studies have shown that NQO1 has been implicated in the pathogenesis of several diseases [43,44,45,46].

## 3. Modulation of Antioxidant Enzyme Expression

In general, oxidative stress is defined as an imbalance between pro-oxidant and antioxidant systems. Excess oxidative stress results in cellular damage due to the oxidation of innumerable essential host macromolecules. Phase II cytoprotective and detoxifying enzymes are responsible for serving as cellular guardians. To date, various signaling pathways, which mediate antioxidant enzyme regulation, have been identified, including protein kinase C (PKC), phosphatidylinositol 3-kinase (PI3K), ER-localized pancreatic endoplasmic reticulum kinase (PERK), mitogen-activated protein kinases (MAPKs), c-Jun NH2-terminal kinase (JNK), AMP-activated protein kinase (AMPK), nuclear factor E2-related factor 2 (Nrf2), activator protein-1 (AP-1), nuclear factor-κB (NF-κB), and cyclic adenosine monophosphate-response element-binding protein (CREB) [47,48,49,50,51,52,53,54,55,56] (Figure 2). 

Numerous cytoprotective genes for detoxifying and antioxidative enzymes in the xenobiotic detoxification and antioxidative response pathways are induced upon exposure to oxidative stress. Nrf2 and the Kelch-like ECH-associated protein 1 (Keap1) system play a central role in antioxidant enzyme regulation [57,58]. Nrf2 is one of the best-characterized antioxidative transcription factors, with an oxidant/electrophile-sensor function. Under resting conditions, Keap1, a cytosolic repressor protein of Nrf2, binds to Nrf2 in the cytosol [59]. Upon exposure to oxidative stress, Nrf2 is separated from the Nrf2-Keap1 complex and translocates into the nucleus, where it induces transcriptional activation of cell defense genes [60]. After translocation to the nucleus, Nrf2 binds to the antioxidant or electrophile response element (ARE/EpRE) in the target gene promoter [61] as do a battery of antioxidant enzymes, including NQO1 [62], HO-1 [63], TR [64], and the Prx [65].

## 4. Effect of Antioxidant Enzyme Expression by Dietary Phytochemicals

Antioxidant enzymes have multifunctional roles in the development and progression of a variety of human diseases, including vascular diseases.

Dietary phytochemicals, which are abundant in fruits and vegetables, have been reported to promote health by enhancing antioxidant and anti-inflammatory abilities, as well as for their capacity for regulating a myriad of signaling mechanisms [66]. Modern scientific approaches have been utilized to identify and study various phytochemicals and have shown the potential value of phytochemicals in the field of pharmacology [67,68,69,70]. Numerous phytochemicals are recognized as inducers of antioxidant enzymes and, therefore, represent attractive candidates for use in healthcare (summarized in Table 1). However, although many studies have been conducted on the physiological activity of phytochemicals in humans, most studies have used animal subjects that have differences in applicable concentrations or metabolism. Therefore, there is an increasing demand for studies into the chemical instability, potential toxicological effects, pharmacokinetics, and pharmacodynamics of the target material to determine the appropriate concentration applicable to humans and to validate the physiological and pharmacological activity.

### 4.1. Anthocyanin

Fruits and vegetables are worthy sources of dietary phytochemicals, such as polyphenols, flavonoids, and carotenoids. In particular, anthocyanins, a class of flavonoids synthesized through the phenylpropanoid pathway, have pronounced antioxidant capacities in vitro and in vivo [102,103]. Anthocyanins stimulate optimal platelet function and exert antithrombotic effects [104]. They have protective effects on visual signal transduction and preclude age-related blindness by reducing the oxidative burden in mouse retinal pigment epithelium (RPE) cells [105]. Huang et al. determined the beneficial effect of anthocyanin on the increase in the levels of the antioxidant enzymes SOD, CAT, and GPx, which might have the potential to be applied to prevent eye diseases such as age-related macular degeneration (AMD) in human retinal capillary endothelial cells [71]. Another study showed that anthocyanins could ameliorate human retinal capillary endothelial function by decreasing ROS and increasing the enzyme activity of CAT and SOD and, therefore, might have the potential to avert the progression of diabetic retinopathy [72]. A recent study indicated that anthocyanins protected rodent endothelial function against high-glucose injury through antioxidant and vasodilatory mechanisms, so anthocyanins could be a promising hypotensive nutraceutical for diabetes [73].

### 4.2. Baicalein

Baicalein (5, 6, 7-trihydroxyflavone) is a natural flavonoid isolated from the root of *Scutellaria baicalensis*, a traditional Chinese herbal medicine commonly used for the treatment of bacterial infections [106]. Baicalein has innumerable biological properties, such as scavenging of relevant toxic ROS in rodent cardiomyocytes [107], and inhibition of tumor-induced angiogenesis [108]. Chao et al. found that baicalein could protect against retinal ischemia in a rat model because of its antioxidant and anti-apoptosis functions and ability to induce HO-1 expression [74]. Shi et al. reported that baicalein ameliorates pathological modifications such as pulmonary arterial remodeling via the MAPK/NF-κB/GPx/SOD pathway in rats [75].

### 4.3. Berberine

Berberine is a well-identified Chinese herbal medicine, extensively used in the treatment of a wide range of inflammatory diseases [109]. Berberine is present in the roots, rhizomes, and stem bulk of plants. Berberine has a variety of biological effects, such as antibacterial, anti-inflammatory, and antioxidant effects, and has been identified as a potential therapeutic candidate for diabetic nephropathy (DN) [110,111,112]. Recent studies have demonstrated the importance of the Nrf2 pathway and expression of the Nrf2-targeted antioxidative genes NQO-1 and HO-1 because of their anti-inflammatory role. These studies suggest that berberine may be beneficial in the development of new therapeutic strategies against inflammatory diseases, such as vascular diseases [76]. Yang et al. suggested that berberine restrained lipid uptake and stimulated cholesterol efflux through Nrf2/HO-1 activation, which resulted in the repression of foam cells and the progression of atherosclerotic plaques in a mouse model [77]. Paul et al. found that berberine has potential as a therapeutic candidate for the treatment of pathologies associated with diabetes in mice [78].

### 4.4. Curcumin

Curcumin (diferuloylmethane) is a polyphenol found in turmeric that has the capacity to treat several chronic diseases related to the cardiovascular system, nervous system, and inflammatory conditions [113,114,115]. Curcumin plays a protective role in the endothelium by inducing HO-1 in bovine aortic endothelial cells (ECs) [116]. Takano et al. found that curcumin suppresses vascular aging and inflammation, which are associated with the elevation of HO-1 in mice [79]. Similarly, Fleenor et al. provided evidence that dietary curcumin supplementation (0.2% in chow) alters two clinically important markers of arterial dysfunction with aging, including normalization of vascular superoxide production and oxidative stress, in a mouse model [80]. Xiao et al. found that curcumin protects against acute vascular inflammation through the activation of the HO-1/Nrf2/ARE/p38 MAPK signaling pathway in rabbits, and thus, may be useful in alleviating the vascular damage that occurs as a result of acute coronary events [81]. The inhibitory effect of curcumin supplementation in high fat diet-induced vascular dysfunction by increasing antioxidant enzyme activities, thereby restraining inflammation and oxidative damage in the vascular endothelium, has been reported in rats [82]. Several studies have reported that the cardioprotective effect of curcumin in a rodent model is associated with the attenuation of oxidant stress [83,84]. Supplementation with curcumin and carotenoids led to the anticipation of low-density lipoprotein (LDL) oxidation, which can be related to an increase in HDL levels and PON1 activity, thereby reducing these cardiovascular risk factors in diabetic rats [85].

### 4.5. Epigallocatechin Gallate (EGCG)

EGCG belongs to the catechin family of polyphenols, and is found in fruits, vegetables, chocolate, wine, and tea [117]. EGCG is the most abundant catechin and is associated with the majority of green tea intake-related health benefits [118]. After traumatic brain injury (TBI), immediate administration of EGCG suppresses edema formation and protect against TBP-induced oxidative stress through the blockage of NADPH oxidase activation in mice [86]. Zheng et al. demonstrated that EGCG provokes Nrf2 and HO-1 via the alteration of caveolae function related to caveolin-1 displacement [87].

### 4.6. Fisetin

Fisetin (3,3,4,7-tetrahydroxyflavone) is a small molecular flavonoid found in many fruits, especially strawberries, and vegetables [119]. Fisetin has been shown to have antioxidative effects mainly through activation of Nrf2/ARE in both rodent and human umbilical vein endothelial cells [88,120,121], and has also been shown to prevent cell proliferation and inflammation [122,123]. Another study revealed that fisetin ameliorated hyperhomocysteinemia (HHcy)-induced endothelial dysfunction and vascular dementia in rats, with the induction of antioxidant genes, such as SOD and CAT [89]. Dong et al. found that fisetin protects against cardiac hypertrophy both in vivo and in vitro, and that repression of oxidative stress is one of the critical underlying mechanisms [90].

### 4.7. Myricetin

Myricetin (3, 5, 7, 3, 4, 5-hexahydroxyflavone) is found in vegetables, fruits, teas, wines, and medicinal plants, and has both anti-inflammatory and antioxidant activities [124,125]. The anti-inflammatory mechanism of myricetin may involve its ability to obstruct the production of pro-inflammatory mediators via the inhibition of the NF-κB, STAT1, Nrf2, and HO-1 pathways [91]. Guo et al. demonstrated that myricetin has a protective effect against oxidative stress in choline-induced vascular dysfunction and liver injury in mice [92].

### 4.8. Quercetin

Quercetin is a bioactive plant flavonol-type flavonoid that is ubiquitous in vegetables and fruits [126]. Numerous human intervention studies have been conducted to evaluate the efficacy of quercetin consumption in reducing the risk of cardiovascular diseases [127,128,129,130,131]. Quercetin has prospective free radical scavenging properties, including inhibition of cancer proliferation [132], neuroprotection [133], renoprotection [134], and anti-thrombosis [135]. In in vitro models using rodent- and human-derived cells. Chis et al. found that quercetin administration, in conjunction with modest exercise training, decreased vascular complications and tissue injuries caused by diabetes in the rat aorta [93]. Likewise, vascular ROS formation and endothelial dysfunction were suppressed by dietary quercetin in HFD-fed ApoE−/− mice, with valuable effects on atherosclerotic plaque formation [94].

### 4.9. Resveratrol

Resveratrol (3,5,4-trihydroxy-trans-stilbene) is a polyphenol found in plants such as peanuts and different types of berries [136,137], which exerts antioxidant, anti-inflammatory, and neuroprotective effects [138,139,140]. Several studies have shown that resveratrol is beneficial for treating and preventing memory deficits in aged rats and has beneficial cardiovascular effects [94,140,141,142]. Resveratrol upregulates the endogenous antioxidant systems, such as the SOD enzymes, in endothelial cells and cardiac myoblasts, and further decreases ROS production [95,143]. A recent study reported that resveratrol ameliorates endothelial dysfunction, memory deficits, increased oxidative stress, inflammation, and impairment of neurotrophin expression in a rat model of vascular dementia [96], and therefore, may be beneficial in DM patients because of its vasculoprotective and neuroprotective effects. In streptozotocin-induced diabetes, resveratrol ameliorates endothelial dysfunction by reducing oxidative stress [97]. Supplementation with resveratrol diminished the presence of atherosclerotic lesions and periarterial fat deposition in ApoE-deficient (apo E−/−) mice [98].

### 4.10. Sulforaphane

Sulforaphane is a natural phytochemical found in cruciferous vegetables, such as broccoli [144]. Sulforaphane is a potent inducer of phase II antioxidant and detoxification enzymes with anticancer, antioxidant, and anti-inflammatory properties [145,146,147]. Sulforaphane plays a protective role in the injury of human cardiovascular cells by lysophosphatidylcholine by preventing the generation of intercellular ROS [99]. Shan et al. found that the action of sulforaphane depends on inflammatory injury in human vascular endothelial cells, mediated by p38 MAPK/JNK, as well as by inducing phase 2 enzymes. [100]. Pretreatment with sulforaphane induces antioxidant defenses in the rat brain and significantly mitigates functional and behavioral deficits after stroke [101].

## 5. Conclusions

Vascular diseases are the leading cause of death worldwide. The underlying mechanisms of vascular disease are many and complex, including excessive generation of reactive oxygen species, oxidative/nitrosative stress, inflammatory responses, and vascular dysfunction. Various aspects of vascular pathology reflect increased oxidative stress, leading to adverse outcomes in the disease state. Mounting evidence indicates that oxidative stress is a major pathological process leading to vascular disease. Increased levels of ROS have been linked to the development and progression of vascular diseases. Cytoprotective antioxidant enzymes are highly effective at diminishing toxicity following exposure to several stressors and inducers, such as ROS.

Phytochemicals are bioactive compounds that are abundantly distributed in fruits and vegetables. The last decade has seen an increase in interest in phytochemicals. Preclinical studies have revealed several beneficial vascular effects of resveratrol, curcumin, and berberine. The advantages appear to be predominantly dependent on the antioxidant, anti-inflammatory, and antithrombotic activities of the compounds. A robust correlation between specific classes of phytochemicals and modulation of antioxidant enzymes was observed.

According to Nelson et al., natural products such as curcumin/curcuminoids, which have various physiological activities, including anti-inflammatory and antioxidant functions, are pharmacologically incompatible in human. They exhibit chemical instability; have poor absorption, distribution, metabolism, excretion, and toxicology properties; and potential toxicological effects in some studies [148]. Therefore, medicinal chemistry research such as pharmacokinetic, pharmacodynamic, and biophysical orthogonal approaches, must be addressed when studying natural bioactive compounds in vitro or in vivo to acquire validated therapeutic efficacy.

This review focuses on understanding the role of antioxidant enzymes in the pathogenesis of vascular diseases. This review has shown that dietary phytochemicals can modulate antioxidant enzyme signaling pathways and, thus, have potential therapeutic value against various chronic diseases, including vascular diseases. A better understanding of the role of antioxidant enzymes by dietary phytochemicals will provide a broader understanding of the vascular system.

## Figures and Tables

**Figure 1 life-11-00199-f001:**
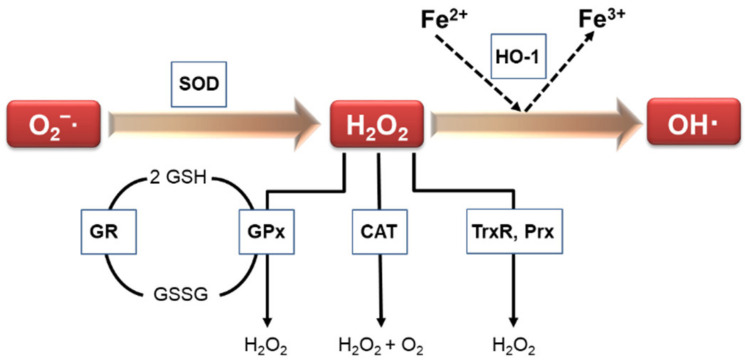
Signaling pathways involved in antioxidant enzymes.

**Figure 2 life-11-00199-f002:**
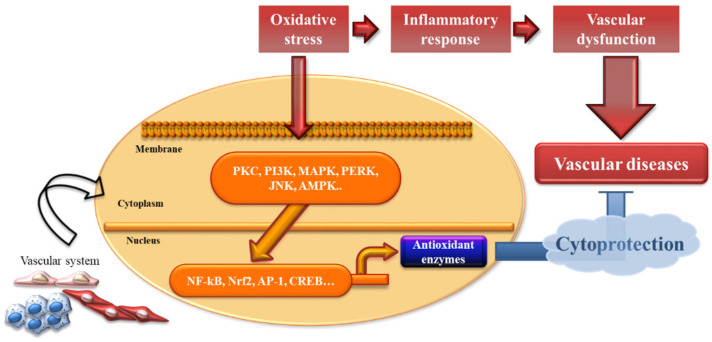
Antioxidant enzymes in vascular protection. Oxidative stress and inflammation in the vascular system response to multiple kinds of inducers, subsequently leading to several pathological conditions. This review supposed that induction of antioxidant enzyme expression provides cytoprotection against vascular injury and preserves vascular function.

**Table 1 life-11-00199-t001:** Summary of phytochemicals with modulating Nrf2 activity and antioxidant enzymes.

Phytochemical	Effects	Altered Antioxidant enzyme
Anthocyanin	Prevent eye disease	SOD, CAT, GPx [71]
	Anti-diabetic effect	CAT, SOD [72]
	Hypotensive effect	HO-1, SOD [73]
Baicalein	Anti-ischemic effect	HO-1 [74]
	Cardiopulmonary protective effect	GPx, SOD [75]
Berberine	Anti-inflammation effect	NQO-1, HO-1 [76]
	Anti-atherosclerotic effect	HO-1 [77]
	Anti-diabetic effect	GR [78]
Curcumin	Anti-atherosclerotic effect	HO-1 [79]
	Cardio-protective effect	SOD [80]
	Anti-inflammation effect	HO-1 [81]
	Vasculoprotective effect	CAT [82]
	Cardio-protective effect	GPx, GR [83]
	Cardio-protective effect	GR, SOD [84]
	Anti-diabetic effect	PON1 [85]
EGCG	Neuroprotective effect	SOD, GPx [86]
	Anti-inflammation effect	HO-1 [87]
Fisetin	Anti-inflammation effect	HO-1 [88]
	Neuroprotective effect	SOD, CAT [89]
	Anti-hypertrophic effect	SOD, CAT, HO-1 [90]
Myricetin	Anti-inflammation effect	HO-1 [91]
	Anti-oxidative effect	SOD, GPx [92]
Quercetin	Anti-diabetic effect	SOD, CAT [93]
	Vasculoprotective effect	HO-1 [94]
Resveratrol	Vasculoprotective effect	SOD, GPx [95]
	Neuroprotective effect	SOD, HO-1 [96]
	Vasculoprotective effect	HO-1 [97]
	Anti-atherosclerotic effect	PON [98]
Sulforaphane	Cardio-protective effect	GPx, SOD, TrxR [99]
	Anti-inflammation effect	TrxR, HO-1 [100]
	Anti-ischemic effect	HO-1 [101]

## Data Availability

Not applicable.

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
