# Peer review of "The Emerging Roles of Antioxidant Enzymes by Dietary Phytochemicals in Vascular Diseases"

_life, 2021, doi:10.3390/life11030199_

Round 1

Reviewer 1 Report

The manuscript presented by Seung Eun Lee and Yong Seek Park review the effect of dietary phytochemicals in their possible role on antioxidant enzymes in vascular disease.

Major concerns:

  1. It would be useful, in line with a published review by Chapple, et al (Signalling networks in focus - Crosstalk between Nrf2 and the proteasome: Therapeutic potential of Nrf2 inducers in vascular disease and aging; The International Journal of Biochemistry & Cell Biology 44 (2012) 1315– 1320) to first make a claim why especially effects of the phytochemicals on antioxidant enzymes is of interest, and not their presumed direct anti-oxidant effects. Since Nrf2 is a dominant factor also described by the authors it would be nice to include this reference to the manuscript.
  2. In this article they refer to Kwak et al. (2003a) to state that besides the compounds SFN and D3T, no other compound showed apparent effects in human subjects. Could authors explain such differences in outcome with regard to all the positive effects shown in the current manuscript?
  3. With regard to the latter remark, with concern to positive effects observed in animal studies there seems to be a discrepancy in the amount given to the animals and the amount proposed for humans. It would be very helpful to discuss this discrepancy and what it means for general supplement intake. This should have a prominent paragraph to help understand the readers to value the contents properly. Also be clear whether a study is conducted in animals or humans and whether concentrations used matched possible concentrations reached in the human situation and whether working concentrations are toxic or not.
  4. How do factors that limit the bioactivity, chemical instability, water solubility, absence of potent and selective target activity, low bioavailability, limit tissue distribution and whether itself is already metabolized before reaching the biologic target affect therapeutic effect of the various phytochemicals?

Textual comments:

  • Rephrase the title: The Emerging Roles of Antioxidant Enzyme Modulation by Dietary Phytochemicals in Vascular disease
  • Page 1 line 16, remove ‘by’
  • Page 1 line 30, remove ‘alteration of’
  • Page 2 line 54, add ‘s’ to neutrophil
  • Page 2 line 82, phrase ‘…that HO-1 in precise...’is not correct
  • Page 4 line 135, phrase ‘…are responsible serve...’is not correct
  • Page 4 line 135, phrase ‘…a various of...’is not correct
  • Page 4 line 153, remove ‘an’ and on line 154 remove ‘s’ from oxidants
  • Page 4 line 157, remove ‘in’
  • Page 5 line 185, add ‘A’ before recent at beginning of sentence
  • Page 6 line 198, remove ‘s’ in pathways
  • Page 6 line 203, add ‘as’ between identifies & potential
  • Page 6 lines 204-206, which studies have demonstrated the importance of the Nrf2 pathway and expression of Nrf2-targeted antioxidative genes? Please add the references. If you refer to refs 83-85 than change ‘In addition, recent’ into ‘These’
  • Page 6, paragraph 4.4 Curcumin. Please address the questions raised by for instance: Nelson KM, et al. The Essential Medicinal Chemistry of Curcumin Mini perspective. J. Med. Chem. 2017, 60, 1620−1637, DOI: 10.1021/acs.jmedchem.6b00975.
  • Page 8, line 298. Add a d to the end of increase.
  • Page 8, 309. Add an s to the end of enzymes.
  • Ref 67, add DOI

Figure 1. Change title into something as: Signalling pathways involved in antioxidant enzymes.

What is multifarious? In line 145

What is meant by “This review propose that due…?

Table 1. Title: Summary of phytochemical inducers of antioxidant enzymes

Author Response

Response to Reviewer comments:

Major concerns:

Comment #1 It would be useful, in line with a published review by Chapple, et al (Signalling networks in focus - Crosstalk between Nrf2 and the proteasome: Therapeutic potential of Nrf2 inducers in vascular disease and aging; The International Journal of Biochemistry & Cell Biology 44 (2012) 1315– 1320) to first make a claim why especially effects of the phytochemicals on antioxidant enzymes is of interest, and not their presumed direct anti-oxidant effects. Since Nrf2 is a dominant factor also described by the authors it would be nice to include this reference to the manuscript.

Answer- We appreciate the reviewer on this comment. Therefore, we have included above reference (#58) on the revised manuscripts following your suggestion.

Comment #2 In this article they refer to Kwak et al. (2003a) to state that besides the compounds SFN and D3T, no other compound showed apparent effects in human subjects. Could authors explain such differences in outcome with regard to all the positive effects shown in the current manuscript?

Answer-  Frankly, many clinical trials investigating antioxidants have been negative. There are some factors that could potentially contribute to the failure of these clinical trials including inadequate knowledge of antioxidant pharmacological actions, insufficient dose-response studies, the presence of interfering drugs which could affect the pharmacokinetics of antioxidants, and the limited sample size. The benefit of antioxidant agents may be based on the oxidative status of each individual. Another factor to take into consideration is the nutritional status of each individual. A balanced diet in high-income countries provides more than sufficient amounts of vitamins, thus further vitamin supplementation is unlikely to confer any benefit and may in fact induce harm. Of notice, however, recent trials have shown a reduced mortality with antioxidant therapies. Such trials, taking into consideration strengths and limitations, do show that oxidative stress plays a pivotal role in CVD and that reduction of oxidative stress reduces cardiovascular and all-cause mortality. These results should encourage scientists to continue to conduct research in the field of oxidative stress and antioxidants. Negative trials about antioxidant agents should not discourage scientists, because in fact, oxidative stress still represents a therapeutic target in CVD. Thus, improvement of experimental settings and knowledge about the pharmacokinetic of phytochemicals, as well as the identification of more specific markers and the use of larger study cohorts, will lead to the identification of novel, more effective therapeutic approaches for vascular disease related to oxidative stress. Therefore, we have included this information with appropriate text modification in the revised manuscript.

Comment #3 With regard to the latter remark, with concern to positive effects observed in animal studies there seems to be a discrepancy in the amount given to the animals and the amount proposed for humans. It would be very helpful to discuss this discrepancy and what it means for general supplement intake. This should have a prominent paragraph to help understand the readers to value the contents properly. Also be clear whether a study is conducted in animals or humans and whether concentrations used matched possible concentrations reached in the human situation and whether working concentrations are toxic or not.

Answer- We really appreciate reviewers for this insight. Accordingly, these informations included in the revised manuscript of paragraph 4 with appropriate text modification (page 5, line 173-179, “However, although many studies have been conducted on the physiological activity of phytochemicals in humans, most studies have used animal subjects which have differences in applicable concentrations or metabolism. Therefore, there is an increasing demand for studies into the chemical instability, potential toxicological effects, pharmacokinetics, and pharmacodynamics of the target material, to determine the appropriate con-centration applicable to humans and the validate the physiological and pharmacological activity.”). Also, we have distinguished all the animal and human models used in references of section 4.

Comment #4 How do factors that limit the bioactivity, chemical instability, water solubility, absence of potent and selective target activity, low bioavailability, limit tissue distribution and whether itself is already metabolized before reaching the biologic target affect therapeutic effect of the various phytochemicals?

Answer- We greatly appreciate the reviewer for this insight. I think this question is the same as the problem raised in paragraph 4.4 below. In order to present the pharmacological basis for the physiological activity and function of natural products, we think that studies such as pharmacokinetic, pharmacodynamic, and biophysical orthogonal for NP should be carried out in terms of medicinal chemistry. Therefore, these informations included in the revised manuscript of conclusion section with appropriate text modification and related reference following your advice. (page 8, line 324-331, “According to Nelson et al., natural products such as curcumin/curcuminoids, which have various physiological activities, including anti-inflammatory and antioxidant functions, are pharmacologically incompatible in human. They exhibit chemical instability, have poor absorption, distribution, metabolism, excretion, and toxicology properties, and potential toxicological effects in some studies [148]. Therefore, medicinal chemistry re-search such as pharmacokinetic, pharmacodynamic, and biophysical orthogonal approaches, must be addressed when studying natural bioactive compounds in vitro or in vivo to acquire validated therapeutic efficacy.”).

Textual comments:

Comment #1 Rephrase the title: The Emerging Roles of Antioxidant Enzyme Modulation by Dietary Phytochemicals in Vascular disease

Answer- We appreciate the reviewer on this comment. We have revised title following your suggestion.

Comment #2 Page 1 line 16, remove ‘by’

Answer- We agree with the reviewer on this comment. We have removed by following your advice.

Comment #3 Page 1 line 30, remove ‘alteration of’

Answer- We appreciate the reviewer on this comment. We have removed alteration of following your advice.

Comment #4 Page 2 line 54, add ‘s’ to neutrophil

Answer- We agree with the reviewer on this comment. We have added s following your advice.

Comment #5 Page 2 line 82, phrase ‘…that HO-1 in precise...’is not correct

Answer- We appreciate the reviewer on this comment. We have removed in precise following your suggestion.

Comment #6 Page 4 line 135, phrase ‘…are responsible serve...’is not correct

Answer- We agree with the reviewer on this comment. We have modified the text following your advice.

Comment #7 Page 4 line 135, phrase ‘…a various of...’is not correct

Answer- We appreciate the reviewer on this comment. We have modified the text following your suggestion.

Comment #8 Page 4 line 153, remove ‘an’ and on line 154 remove ‘s’ from oxidants

Answer- We appreciate the reviewer on this comment. We have removed an and s following your advice.

Comment #9 Page 4 line 157, remove ‘in’

Answer- We agree with the reviewer on this comment. We have removed in following your advice.

Comment #10 Page 5 line 185, add ‘A’ before recent at beginning of sentence

Answer- We appreciate the reviewer on this comment. We have modified the text following your suggestion.

Comment #11 Page 6 line 198, remove ‘s’ in pathways

Answer- Thanks for your comment. We have removed s following your advice.

Comment #12 Page 6 line 203, add ‘as’ between identifies & potential

Answer- We agree with the reviewer on this comment. We have modified the text following your suggestion.

Comment #13 Page 6 lines 204-206, which studies have demonstrated the importance of the Nrf2 pathway and expression of Nrf2-targeted antioxidative genes? Please add the references. If you refer to refs 83-85 than change ‘In addition, recent’ into ‘These’

Answer- We greatly appreciate the reviewer for this insight. Additional recent studies in the text refer to following refs 86-88, therefore we revised text on line 206-207 to “These studies suggest”.

Comment #14 Page 6, paragraph 4.4 Curcumin. Please address the questions raised by for instance: Nelson KM, et al. The Essential Medicinal Chemistry of Curcumin Mini perspective. J. Med. Chem. 2017, 60, 1620−1637, DOI: 10.1021/acs.jmedchem.6b00975.

Answer- We really appreciate reviewers for this insight. In terms of medicinal chemistry, analysis of curcumin using the NAPRALERT database showed unexpected and interesting results. This study discusses that curcumin is a PAINS, IMPS, and Poor Lead Compound, and it is an unstable and highly reactive compound, so a new evaluation of curcuminoids is needed based on in-depth evaluation. In addition, they supposed that curcumin is a compound with promiscuous biological activity similar to ginsenosides Rb-1, Rg-1, genistein, quercetin, apigenin, nordihydroguaiaretic acid, resveratrol, kaempferol, and fisetin which are pharmacological research should be supplemented. However, various biological studies showing effective anti-inflammatory, anti-cancer, anti-oxidant, anti-viral properties of flavonoids or phytochemicals and other natural products have been verified. Therefore, not only studies on the physiological activity of natural products such as cucrcumin or fisetin, but also medicinal chemistry research for the correct extraction, purification, detection, pharmacokinetic/pharmacodynamic/biophysical orthogonal assay are must be followed from now on further study together. Accordingly, these informations included in the revised manuscript of conclusion section with appropriate text modification with Ref number 148 (page 8, line 324-331).

Comment #15 Page 8, line 298. Add a d to the end of increase.

Answer- We appreciate the reviewer on this comment. We have modified text following your suggestion.

Comment #16 Page 8, 309. Add an s to the end of enzymes.

Answer- We agree with the reviewer on this comment. We have revised text following your advice.

Comment #17 Ref 67, add DOI

Answer- We appreciate the reviewer on this comment. We have revised Ref 67 following your advice.

Comment #18 Figure 1. Change title into something as: Signalling pathways involved in antioxidant enzymes.

Answer- Thank you for your kind advice. We have modified title of figure 1 following your advice.

Comment #19 What is multifarious? In line 145

Answer- We appreciate the reviewer on this comment. We have modified text following your suggestion.

Comment #20 What is meant by “This review propose that due…?

Answer- We appreciate the reviewer on this comment. We have revised all the manuscripts using standard and definite English language usage following your suggestion also, attached certificate to cover letter.

Comment #21 Table 1. Title: Summary of phytochemical inducers of antioxidant enzymes

Answer- We appreciate the reviewer on this comment. We have modified title of table 1 following your suggestion.

Reviewer 2 Report

Dear authors,
Your manuscript provides an interesting source of information about modulation of antioxidant enzymes by dietary phytochemicals in vascular diseases. The manuscript is processed in a high-quality, clear, and well-arranged form that has a logical structure and the individual parts are connected.

Firstly, I have questions for the authors. You are writing in the manuscript (page 4): “Numerous phytochemicals are recognized inducers of antioxidant enzymes and therefore represent attractive candidates for use in healthcare (summarized in Table 1)“. What criteria were chosen for the selection of these phytochemicals in Table 1? Why the compounds listed in the Table 1 were chosen? This fact is not specified in the manuscript. Are they dietary phytochemicals, as the title of the manuscript mentions? Nevertheless, berberine and baicalein are not a common part of the human diet.  There are many other dietary phytochemicals affecting antioxidant enzymes in connection with cardiovascular diseases (other flavonoids, phenolic acids, lycopene etc.).

Secondly, I have a few minor comments:

p. 5 line 176 in vivo must be written in italic letters

I recommend changing legend of Table 1: Summary of phytochemical and antioxidant enzyme. I think, it would be more appropriate for example: Selected phytochemicals with modulating activity on antioxidant enzymes.

Two Figures 1. The Figure 1 on page 4 is Figure 2.

Conclusions: The last paragraph - different font size without indenting the first line of paragraph

References section needs revision. Homogenize the writing of journals titles.  For example: reference 3. Oxid Med Cell Longev 2017; reference 9. Arteriosclerosis Thrombosis and Vascular Biology 2011; references 10. Arterioscler. Thromb. Vasc. Biol. Med. 2013

Reference 9. wrong page numbers, 3011-U3619 instead of 3011-3019

p. 9 line 362 10.1016/j.bbrc.2010.03.083. I do not understand what the number belongs to. The same discrepancy is repeated p. 12 line 468; p. 13 lines 501, 519, 523, 539; p. 14 lines 548, 565, 576; p. 15 lines 584, 591,606,

I consider that this manuscript can be accepted with minor revision.

Author Response

Response to Reviewer comments:

Dear authors, your manuscript provides an interesting source of information about modulation of antioxidant enzymes by dietary phytochemicals in vascular diseases. The manuscript is processed in a high-quality, clear, and well-arranged form that has a logical structure and the individual parts are connected.

Comment #1 Firstly, I have questions for the authors. You are writing in the manuscript (page 4): “Numerous phytochemicals are recognized inducers of antioxidant enzymes and therefore represent attractive candidates for use in healthcare (summarized in Table 1)“. What criteria were chosen for the selection of these phytochemicals in Table 1? Why the compounds listed in the Table 1 were chosen? This fact is not specified in the manuscript. Are they dietary phytochemicals, as the title of the manuscript mentions? Nevertheless, berberine and baicalein are not a common part of the human diet.  There are many other dietary phytochemicals affecting antioxidant enzymes in connection with cardiovascular diseases (other flavonoids, phenolic acids, lycopene etc.).

Answer- We really appreciate reviewers for this insight. This table is based on phytochemicals that activate the Nrf2 signaling pathway and thereby modulate the activity of antioxidant enzymes involving NQO1, HO-1, and TR, as written in the text. Therefore, we have modified the text in legend of table 1 following your advice.

Minor comments:

Comment #1 p. 5 line 176 in vivo must be written in italic letters

Answer- We appreciate the reviewer on this comment. We have modified text following your suggestion.

Comment #2 I recommend changing legend of Table 1: Summary of phytochemical and antioxidant enzyme. I think, it would be more appropriate for example: Selected phytochemicals with modulating activity on antioxidant enzymes.

Answer- We agree with the reviewer on this comment. We have modified text in the legend of table 1 following your suggestion.

Comment #3 Two Figures 1. The Figure 1 on page 4 is Figure 2.

Answer- We appreciate the reviewer on this comment. We have revised text following your advice.

Comment #4 Conclusions: The last paragraph - different font size without indenting the first line of paragraph

Answer- Thank you for your kind comment. We have revised font size following your suggestion.

Comment #5 References section needs revision. Homogenize the writing of journals titles.  For example: reference 3. Oxid Med Cell Longev 2017; reference 9. Arteriosclerosis Thrombosis and Vascular Biology 2011; references 10. Arterioscler. Thromb. Vasc. Biol. Med. 2013

Answer- We appreciate the reviewer on this comment. We have revised reference style following your advice.

Comment #6 Reference 9. wrong page numbers, 3011-U3619 instead of 3011-3019

Answer- We agree with the reviewer on this comment. We have modified reference style following your advice.

Comment #7 p. 9 line 362 10.1016/j.bbrc.2010.03.083. I do not understand what the number belongs to. The same discrepancy is repeated p. 12 line 468; p. 13 lines 501, 519, 523, 539; p. 14 lines 548, 565, 576; p. 15 lines 584, 591,606,

Answer- We greatly appreciate the reviewer for this comment. We have revised and double-checked all the references using EndNote program with most recently updated reference style following your suggestion.

Reviewer 3 Report

I have read the manuscript with interest. The topic falls in the scope of this journal. The design is reasonable and the authors have given good number of citations about the subject, but some parts should be expanded. However, there are still some issues that have to be addressed by the authors before considering the manuscript for publication. My comments are detailed below.

Title: please standardize the title by also changing “enzymes” and “diseases” with the first capital letter.

Keywords: It is better to avoid repetition of words already present in the title. For example, I suggest to replace “phytochemicals” with “bioactive compounds”.

Introduction: I suggest to expand this section.

Table 1: I suggest to add 2 columns, one with the potential therapeutic benefit of phytochemicals in disease (i.e. named “Effects”) and the other with “References”.

Conclusions: Line 311-316 change the font size.

References: Fix up and align the “References” (e.g. line 361-362, 467-468,500-501, etc.)

Author Response

Response to Reviewer comments:

I have read the manuscript with interest. The topic falls in the scope of this journal. The design is reasonable and the authors have given good number of citations about the subject, but some parts should be expanded. However, there are still some issues that have to be addressed by the authors before considering the manuscript for publication. My comments are detailed below.

Comment #1 Title: please standardize the title by also changing “enzymes” and “diseases” with the first capital letter.

Answer- We appreciate the reviewer on this comment. Accordingly, we have modified the text in title following your suggestion.

Comment #2 Keywords: It is better to avoid repetition of words already present in the title. For example, I suggest to replace “phytochemicals” with “bioactive compounds”.

Answer- We really appreciate reviewers for this insight. Therefore, we have modified the text following your advice.

Comment #3 Introduction: I suggest to expand this section.

Answer- We greatly appreciate the reviewer for this insight. Therefore, we have modified the text in introduction. (page 1, line 31-35, “The oxidative balance in the vasculature is tightly regulated by a wealth of pro- and anti-oxidant systems that contain antioxidant enzymes. A better understanding of oxidative stress and modulation of antioxidant enzymes is necessary for the broader use of pharmacological and regenerative therapies for vascular diseases.”)

Comment #4 Table 1: I suggest to add 2 columns, one with the potential therapeutic benefit of phytochemicals in disease (i.e. named “Effects”) and the other with “References”.

Answer- We appreciate the reviewer on this comment. Therefore, we have modified the text and table 1 following your suggestion.

Comment #5 Conclusions: Line 311-316 change the font size.

Answer- Thank you for your kind comment. Accordingly, we have modified the font size following your advice.

Comment #6 References: Fix up and align the “References” (e.g. line 361-362, 467-468,500-501, etc.)

Answer- We appreciate the reviewer on this comment. Accordingly, we have revised and double-checked all the references following your advice.

Round 2

Reviewer 1 Report

no comments